# Yeast Protein Asf1 Possesses Modulating Activity towards Protein Kinase CK2

**DOI:** 10.3390/ijms232415764

**Published:** 2022-12-12

**Authors:** Andrea Baier, Ryszard Szyszka, Monika Elżbieta Jach

**Affiliations:** 1Department of Animal Physiology and Toxicology, Faculty of Medicine, The John Paul II Catholic University of Lublin, Konstantynów 1i, 20-708 Lublin, Poland; 2Department of Molecular Biology, Faculty of Medicine, The John Paul II Catholic University of Lublin, Konstantynów 1i, 20-708 Lublin, Poland

**Keywords:** protein kinase CK2, phosphorylation, enzyme inhibition, yeast protein Asf1

## Abstract

Protein kinase CK2 plays an important role in cell survival and protects regulatory proteins from caspase-mediated degradation during apoptosis. The consensus sequence of proteins phosphorylated by CK2 contains a cluster of acidic amino acids around the phosphorylation site. The poly-acidic sequence in yeast protein Asf1 is similar to the acidic loop in CK2β, which possesses a regulatory function. We observed that the overexpression of Asf1 in yeast cells influences cell growth. Experiments performed in vitro and in vivo indicate that yeast protein Asf1 inhibits protein kinase CK2. Our data suggest that each CK2 isoform might be regulated in a different way. Deletion of the amino or carboxyl end of Asf1 reveals that the acidic cluster close to the C-terminus is responsible for the activation or inhibition of CK2 activity.

## 1. Introduction

Protein kinase CK2 is a constitutively active tetrameric enzyme assigned to the CMGC group of protein kinases that is independent of second messengers and is regulated through protein–protein interactions, changes in its concentration and localization, and oligomerization [1,2]. CK2-affecting phosphoacceptor sites of proteins are located at serine/threonine residues that are surrounded by numerous acidic residues (either Asp/Glu or phosphorylated amino acids); amongst them, the one at position n + 3 plays the most important role [3], although, in a few cases, the phosphorylation of tyrosine residues has been already described [4,5]. It has been estimated that CK2 might be responsible for the generation of a substantial part of the eukaryotic phosphoproteome [6]. CK2 is involved in many cellular processes, such as gene expression, cell growth and differentiation, embryogenesis and apoptosis, circadian rhythms, and cell cycle regulation [7,8,9,10,11]. It has been proposed that CK2 plays an important role in the transduction of survival signals and protect the cell against stress [1,9,10,12]. Abnormal levels and nuclear localization of CK2 were originally found in rapidly proliferating cells in several types of cancer, including lung, kidney, prostate, mammary gland, and head and neck cancer. It has become apparent that CK2 is dysregulated by an increase in protein expression in all cancer tissues examined so far [13,14]. CK2 is a modulator of all the hallmarks of tumor. It promotes oncogenesis and creates a cellular environment favorable to neoplasia [15]. These findings make CK2 a potentially important target for cancer therapy [13,16,17,18]. CK2 also influences inflammatory, neurodegenerative (e.g., Alzheimer’s and Parkinson’s diseases), vascular, bone tissue, and skeletal muscle disorders, and viral diseases, including COVID-19 [19,20].

Human protein kinase CK2 is a tetramer composed of two catalytic α or α’ subunits with molecular weights of 42 kDa and 38 kDa, respectively, which bind to a dimer of the 27 kDa β subunit with a stoichiometry α_2_β_2_, α’_2_β_2_, or αα’β_2_ [1,2,21]. The heterotetramer can dissociate under certain conditions, indicating the existence of individual subunits [22]. CK2α and CK2α’ are homologous and differ only in the C-terminal region, while the β subunit lacks homology to any other regulatory proteins [23]. The regulatory subunit CK2β is not essential for activity but can affect the ability of the catalytic subunits to phosphorylate certain substrates [24]. The free catalytic α/α’ subunits hold their activity against a range of substrates, while the regulatory β subunits might exhibit functions independent of CK2, targeting the cell cycle, DNA repair, and protein kinases such as Chk1, A-Raf, and c-Mos [25].

The CK2α subunit is described to be expressed ubiquitously in all tissues [2], whereas the CK2α’ subunit can be mainly found in the brain and testicles, supporting the notion that this subunit may have specific functions [26]. Indeed, knockout of the CK2α’ subunit in mice showed a phenotype similar to globozoospermia in humans, whereas knockout of the CK2α and CK2β subunits was embryonically lethal [27,28,29]. Genetic studies performed on baker’s yeast have shown that the CK2α’ subunit is involved in cell cycle progression, while the second one (CK2α) is required for the maintenance of cell polarity [30]. Disruption of yeast CK2β does not affect yeast viability. In yeast cells lacking the CK2 holoenzymes (double deletion mutant), the binding of the SBF and MBF complexes to G_1_ promotors is disturbed, and, therefore, the S-phase entrance is delayed [31]. Interestingly, in human osteosarcoma cells with tetracycline-regulated CK2 expression, only the expression of inactive CK2α’ was induced, and not that of inactive CK2α; the significantly decreased cell proliferation and viability give additional evidence for the functional specialization of CK2 isoforms [32].

Many studies regarding CK2’s regulation and function have been focused on the identification of cellular substrates and interacting proteins. Based on the high sequence homology between the two catalytic subunits of human CK2, it is likely that many of the proteins that have been found in complex with CK2 will interact in similar way with both the α and α’ subunits. Olsen and Guerra [33] have listed almost 70 proteins that interact with CK2, and, for some of the interacting proteins, an effect on CK2 activity has been shown [34].

The identification of protein–protein interactions very often provides insights into its function. The acidic surrounding of the recognition site was used to query a baker’s yeast database (SGD) for proteins that may have a potential role in the regulation of apoptosis and/or CK2 activity. From a few potential candidates, the nucleosome assembly factor Asf1, which possesses an over 20-amino-acid-long C-terminal cluster of aspartic and glutamic acids, was chosen as a potential CK2-modulating protein. A global analysis of protein complexes in *Saccharomyces cerevisiae* has shown that all four subunits of yeast protein kinase CK2 form complexes with many proteins, including with Asf1 [35].

The Asf1 protein (a homolog to human CIA) is a ubiquitous eukaryotic histone-binding, transcription-regulatory, and deposition protein that mediates nucleosome formation in vitro and is required for genome stability in vivo [36]. Asf1 acts as a histone chaperone during DNA replication through specific interactions with histone H3/H4 and the histone deposition factor CAF-I [36,37]. Its roles in replication, chromatin remodeling, and DNA repair have also established Asf1 as an important component of a number of chromatin assembly and modulation complexes. In addition to its role in conserved interactions with proteins involved in chromatin silencing, transcription, chromatin remodeling, and DNA repair [36], Asf1 is involved in the induction of apoptosis [38]. Moreover, one of the human paralogs, Asf1B, is overproduced in most cancer tissues in comparison with normal ones. Therefore, Asf1B may be recognized as an independent prognostic agent in multiple types of cancer [39].

The yeast Asf1 protein differs from its animal homolog proteins and contains (1) an N-terminal 155-amino-acid-long sequence with high homology (65%-90%) to other eukaryotic species and (2) a long poly-acidic tract present in fungi (in baker’s yeast, an over 20-amino-acid-long cluster of aspartic and glutamic acids) [36,40]. This pseudoinhibitory acidic cluster may influence CK2’s antiapoptotic activity in the eukaryotic cell, which is observed as the induction of apoptosis in the case of the overexpression of yeast Asf1 [41,42].

The present paper focuses on the analysis of the influence of the Asf1 protein on human and yeast protein kinase CK2 isoforms.

## 2. Results

The recombinant proteins, human CK2α and CK2α’ subunits, CK2β, and the three Asf1 protein variants were overexpressed in *E. coli* cells, whereas yeast CK2s were expressed in *S. cerevisiae* cells and purified as described in the Materials and Methods section.

The acidic surroundings of the recognition site in proteins phosphorylated by CK2 were used to search for potential protein partners of CK2. As a result of the *Saccharomyces* genome database search, we identified yeast protein Asf1. The acidic sequence of yeast Asf1 protein (i.e., **D^170^DEEEEDDEEEDDDEDDEDDEDDD^193^**) shows clear similarity to the CK2 consensus sequence **S/T**XXE/D and high similarity to the acidic loop of the human regulatory CK2β subunit (D^55^LEPDEELED^64^), as shown in Figure 1. Based on its similarity to the clusters of amino acids that are typically present in CK2 substrates, this region is reminiscent of autoinhibitory sequences identified in a large number of other protein kinases.

In our previous study, we showed that Asf1’s interaction with protein kinases promoted the negative regulation of this enzyme in vivo [43]. Firstly, a possible effect of the yeast Asf1 protein was examined using the recombinant yeast Asf1 protein, as well as the recombinant human and yeast catalytic subunits CK2α and CK2α’ (Figure 2). Asf1 exhibits inhibitory effects towards both subunits. The phosphorylation reaction was carried out using a constant concentration of recombinant yeast acidic ribosomal P2B as a protein substrate (10 µM), 20 µM ATP as a phosphate donor, and increasing concentrations of Asf1 (0–12 µM). Interestingly, as in the case of other modulators, such as heparin, polylysine, NaCl, or some inhibitors, the influence differed between the two subunits. The α subunit was statistically significantly activated in the presence of a low concentration of Asf1 (0.2 µM, *p* < 0.05), and, at higher concentrations, it was inhibited. Compared with the control, the α’ subunit was significantly inhibited (*p* < 0.01) compared to the α subunit, with IC_50_ values of 1.3 and 2.5 µM, respectively, under standard reaction conditions.

Next, we examined how the Asf1 protein affects the individual isoforms of human and yeast CK2 (Figure 3). The reaction mixture was incubated for 30 min with a 15 min preincubation step of CK2 and Asf1 protein. In the case of the free catalytic CK2α’ subunit, we noted a significant inhibitory effect *p* < 0.01) compared to that under standard conditions, while, in the case of the catalytic α subunit or the holoenzyme α_2_β_2_, we observed a significant stimulatory effect (*p* < 0.05) compared to the control. This could indicate that the Asf1 protein interacts in different way with each CK2 isoform. It is not clear what causes the differences in the inhibitory effects of Asf1 with various molecular forms of CK2, but the results might suggest various protein–protein interactions (especially in the case of holoenzymes) and differences in regulatory mechanisms.

As already described in our former studies [44,45,46,47], synthetic and natural inhibitors have different inhibitory potential depending on the protein substrate. We have investigated the inhibitory action of the Asf1 protein using already characterized substrates from yeast *S. cerevisiae*, such as recombinant Svf1, Fip1, and Elf1 [48,49,50]. The experiments using human catalytic subunits showed that the Asf1 protein exhibited an inhibitory effect towards the phosphorylation of different substrates. This could indicate that the Asf1 protein suppresses the enzymatic activity of CK2α’ for its various substrates. Nevertheless, the intensity of inhibition differs, as shown in Figure 4. Whereas the phosphorylation of P2B and Fip1 is significantly inhibited (*p* < 0.01), with IC_50_ values of 1.3 and 1.6 µM in the case of CK2α’, respectively, the phosphorylation of the synthetic peptide RRRADDSDDDDD was not significantly different from that of the control at 10 µM. Svf1 and Elf1 phosphorylation was significantly inhibited (*p* < 0.05), with inhibition constants of 9.8 and 6.0 µM. Similar results were obtained for the CK2α subunit but with lower IC_50_ values.

It is tempting to speculate that the potential fragment responsible for the modulation of CK2 is located in the C-terminus of the yeast Asf1 protein. We wondered whether the whole Asf1 molecule is necessary for CK2 inhibition, or whether only a specific part within the protein is needed. Based on this, the effect of trypsin-digested full-length Asf1 towards CK2 activity was examined. Due to the action of trypsin, two longer and several short peptides were obtained (Figure 5A). Additionally, two shorter Asf1 constructs were tested, comprising either the conserved amino acids 1-169 or amino acids 170-279, including the characteristically acidic stretch (Figure 5A). The phosphorylating activities of human and yeast CK2 were analyzed under standard conditions in the presence of 5 µM of Asf1 protein or its fragments. As shown in Figure 5B,C, the obtained results suggest that the C-terminal part (aa 170-279) is responsible for the decrease in CK2 activity. The results of trypsin-digested Asf1 support this claim. The catalytic activity of CK2α’ was significantly inhibited by the complete Asf1 protein (approximately 70–75%, *p* < 0.01). A slightly lower level (around 8–10%) of CK2α’ inhibition was observed when peptides obtained after hydrolysis by trypsin were used instead of the full-length Asf1 protein. The Asf1 molecule lacking the C-terminal acidic stretch Asf1^1-169^ has no CK2-activity-inhibiting properties. This supports the thesis that the segment responsible for the inhibitory effect on the catalytic activity is the characteristic acidic fragment of the yeast Asf1 protein, due to the fact that its removal from the Asf1 protein restored the full kinase activity of the catalytic subunit.

The C-terminal acidic fragment of yeast Asf1 is similar to both the pseudo-substrate region of the regulatory CK2β subunit and the recognition sequence present in proteins modified by CK2. Therefore, a possible mechanism of inhibition could involve competition for the protein-substrate-binding site. We wondered whether the increase in the protein substrate concentration in the reaction mixture affected the inhibitory potential of Asf1. Changing the standard reaction conditions yielded different results for the α and α’ subunits. Inhibition studies were performed at fixed concentrations of protein substrate (P2B) and at variable concentrations of ATP in the absence or in the presence of increasing concentrations of inhibitor. The inhibition of both catalytic subunits in the presence of 5, 10, or 20 μM ATP revealed another mode of action between the subunits. The inhibitory effect estimated in the case of CK2α’ was similar to yet independent of the ATP concentration. On the other hand, the CK2α subunit was more inhibited in the presence of 5 µM ATP than 20 µM, suggesting an ATP-dependent inhibition mode (Figure 6A,B). Applying different protein substrate (P2B) concentrations (25, 50, 75 μM) also led to various effects. The extent of inhibition was changed only in the case of the CK2α’ subunit (Figure 6C,D). The phosphorylating activity was less inhibited in the case of higher substrate concentrations, suggesting a competitive inhibition mode towards the protein substrate. The experiments were performed with the full-length Asf1 protein and the C-terminal fragment Asf1^170-279^. As shown in Figure 6, the inhibition mode is the same, independent of the used Asf1 molecule.

The data suggest that the interaction of the kinase with the inhibiting Asf1 protein is not exclusively mediated by pseudo-substrate-reproducing motifs. Amino acid stretches lying beyond the catalytic center of the kinase and/or beyond the pseudo-substrate region of the inhibiting protein could influence the extent of inhibition. To address this question, we analyzed the interaction of both CK2 catalytic subunits with the acidic stretch of Asf1 comprising aa 170-193 (DDEEEEDDEEEDDDEDDEDDEDDD) using the CABS-dock method. As shown in Figure 7 and Table 1, the binding of the peptide to the enzyme differs between CK2α and CK2α’. In the case of CK2α’, the peptide stretches from the ATP-binding pocket (K^50^) to the activation loop (S^195^). Both sites are responsible for the enzyme activity and the substrate recognition. In the case of CK2α, the peptide binds to the catalytic loop (K^158^) and activation loop (E^180^), but not to the ATP-binding pocket. For this reason, we predict that this could be one explanation for the higher extent of inhibition in the case of CK2α’ when compared with CK2α; as shown in this work, both subunits are differently inhibited, dependent on the protein substrate. This might be explained by the findings of the interaction with the substrate recognition site. The donor–acceptor distances are between 2.7 and 3.5 Å, which are characterized as moderate, typically found in proteins [51].

In a previous study, the effects of TBB and TBI on the cell growth of several yeast strains were analyzed [46]. TBB decreased the cell growth of both mutants lacking either the CK2α or CK2α’ subunit. In the case of TBI, no significant inhibition could be observed.

The first in vivo experiments revealed that the effect of the Asf1 protein towards protein kinase CK2 is also detectable in yeast cells. In further in vivo experiments, conducted on Asf1-transformed *S. cerevisiae* cells, CK2 activity was analyzed. Lysates from yeast overexpressing Asf1 possessed lower phosphorylating activity than lysates from non-transformed cells [43]. These in vivo results confirmed the results from enzymatic activity studies of yeast CK2. In further experiments, we estimated the growth rate of yeast cells to measure the influence of Asf1 overexpression on cell division. For this purpose, *S. cerevisiae* strains (wild-type and deletion mutants) were grown either in the presence or absence of overexpressed Asf1 protein (Figure 8A,B).

The experiments were performed in two different media. One contained glucose as a carbon source, and the second contained galactose, which also induced the overexpression of the Asf1 protein. The growth of the wild-type BMA64-1A was inhibited when Asf1 was overexpressed. Interestingly, the cell division of yeast expressing only the CKα’ subunit (Y01428) was inhibited more strongly compared to the wild-type BMA64-1A. This divergence is explainable by the fact that yeast cells lacking the CK2α’ subunit (Y01837) grew much faster. In the case of the basal expression of Asf1 (medium with glucose), this deletion mutant exhibits more than two-times faster growth than the control strain. Experiments carried out expressing the shorter Asf1^170-279^ protein gave results similar to those with the full-length Asf1 protein. After 8 h cultivation, cell lysates were analyzed towards their expression levels of His-tagged Asf1 protein (Figure 8C,D). Yeast cells grown in medium supplemented with 2% galactose overexpressed Asf1 when compared to the basal expression of Asf1 in cells cultured in medium containing 2% glucose. This difference in the expression of Asf1 resulted in altered effects on the growth rate.

## 3. Materials and Methods

### 3.1. Materials

*Saccharomyces cerevisiae* strains BMA64-1A (wild-type), Y01837 (CK2α’ deletion mutant, ΔCK2α), Y01428 (CK2α deletion mutant, ΔCK2α’) were obtained from EUROSCARF (Scientific Research and Development GmbH, Oberursel, Germany). Restriction enzymes were purchased from Thermo Fisher Scientific Polska (Warsaw, Poland). All other materials were purchased from Merck KGaA (Darmstadt, Germany) if not otherwise stated.

### 3.2. Purification of Yeast CK2

Yeast strains *Saccharomyces cerevisiae* were cultivated under aerobic conditions in YPD medium, i.e., yeast extract, peptone, glucose (Oxoid Ltd. Hampshire, UK) at 28 °C to the exponential growth phase. Yeast cells were collected by centrifugation (4000× *g*) for 5 min.

Yeast CK2 holoenzymes with the compositions of αα’ββ’, α_2_ββ’, and α’_2_ββ’ were obtained from yeast strains BMA64-1, Y01837, and Y01428, respectively. Native CK2 holoenzymes were isolated and purified from yeast ribosomal-free extract as previously described [52], using different chromatography steps, such as ion-exchange chromatography, i.e., P11- and DE52-cellulose, followed by affinity chromatography, i.e., α-casein and heparin sepharose.

Recombinant free catalytic subunits CK2α and CK2α’ were expressed and purified as previously described [53].

### 3.3. Overexpression and Purification of Human CK2α and CK2α’ Subunits

Both catalytic subunits were overexpressed and purified as described previously [44]. Briefly, *E. coli BL21*(*DE3*)*trxB* cells (Novagen Merck KGaA, Darmstadt, Germany) harboring the plasmid pGEX-3X::*hsCK2α* or pGEX-3X::*hsCK2α*’ were grown until OD_600_ = 0.6 at 37 °C. Next, IPTG was added to the final concentration of 0.2 mM and the cultures were continued at room temperature for 4 h and then centrifuged at 5000× *g* for 10 min. Bacterial cells were disrupted by sonication and the supernatant was purified using glutathione-sepharose (Amersham Pharmacia Biotech UK Ltd., Buckinghamshire, UK). Fractions containing the CK2 subunit were pooled and dialyzed against 50 mM Tris/HCl buffer, pH 7.5, supplemented with 6 mM β-mercaptoethanol and 30% glycerol. The obtained protein preparations were used in enzymatic assays.

### 3.4. Purification of Human CK2 Holoenzymes

CK2α_2_β_2_ and CK2α’_2_β_2_ holoenzymes were purified as described by Turowec et al. [54]. Briefly, both catalytic subunits were overexpressed as GST fusion proteins, whereas the regulatory subunit was expressed as a His-tagged protein. Obtained bacterial pellets were mixed prior to lysis. Lysates were purified as previously described for free catalytic subunits using glutathione-sepharose.

### 3.5. Cloning of the Asf1, Asf1^1-169^, and Asf1^170-279^ Proteins into Bacterial Expression Vector pET28a

In order to overexpress the yeast Asf1 protein and its N- and C-terminal fragments, the appropriate nucleotide sequences were cloned into the pET28a vector (Novagen Merck KGaA, Darmstadt, Germany) using yeast genomic DNA as a template for standard PCR and primers adding the *BamHI* and *SalI* restriction sites, as listed in Table 2.

### 3.6. Cloning of Asf1 Protein and the C-Terminal Fragment Asf1^170-279^ into Yeast Expression Vector pYES2/CT

In order to overexpress the Asf1 protein and its C-terminal fragment aa 170-279 in *S. cerevisiae*, the appropriate nucleotide sequences were cloned into the pYES2/CT vector (Invitrogen Thermo Fisher Scientific Polska, Warsaw, Poland) using the pET28a::*asf1* plasmid as a template for standard PCR and primers adding *HindI* and *EcoRI* restriction sites for Asf1 and Asf1^170-279^, as listed in Table 2.

### 3.7. Overexpression and Purification of Asf1, Asf1^1-169^, and Asf1^170-279^ Proteins

*E. coli* cells harboring the pET28a::*asf1*, pET28a::*asf1*(*1-169*), or pET28a::*asf1*(*170-279*) plasmid were cultured in terrific broth medium (Novagen Merck KGaA, Darmstadt, Germany) until they reached an OD_600_ = 0.6. IPTG was added to a final concentration of 0.5 mM and the culture was further incubated at 37 °C for 4 h. The cells were harvested by centrifugation at 5000× *g* for 5 min. The cell lysate was purified by affinity chromatography using Ni-NTA.

### 3.8. Protein Substrates

The protein substrates P2B, Svf1, Fip1, and Elf1, used as phosphate acceptors, were overexpressed and purified as previously described [44,46,47,48]. The acidic peptide with the sequence RRRADDSDDDDD was purchased from Sigma-Aldrich Merck KGaA, Darmstadt, Germany.

### 3.9. Protein Phosphorylation

CK2 activity was determined in a standard reaction mixture (40 μL of final volume) containing 20 mM Tris–HCl buffer, pH 7.5, 15 mM MgCl_2_, 6 mM β-mercaptoethanol, and 20 μM [γ-^32^P]ATP (specific radioactivity 500–1000 cpm/pmol, Hartmann Analytic GmbH, Braunschweig, Germany) in the presence of 10 µM protein substrate (P2B, Elf1, Svf1, Fip1). Incubation was performed at 37 °C for 15 min. Afterwards, the reaction was terminated by adding 7 μL of the SDS-PAGE loading buffer. Reaction mixtures were resolved in SDS-PAGE, followed by Coomassie blue staining and autoradiography. The phosphate incorporation level in the protein was estimated by cutting off the corresponding band and measuring the radioactivity in a scintillation counter (PerkinElmer, Waltham, USA).

Phosphorylation activity using the synthetic peptide RRRADDSDDDDD (50 µM) was analyzed by spotting an aliquot of the reaction mixture onto P81-cellulose paper (Whatman International Ltd., Kent, UK). Next, filters were washed thrice with 1% phosphoric acid and dried before counting in a scintillation counter.

### 3.10. Asf1 Digestion by Trypsin

First, 100 μg of full-length Asf1 protein was digested by 2 μg of trypsin. The reaction was carried out for 18 h at 37 °C. Afterwards, the reaction was stopped by adding soybean trypsin inhibitor (STI).

### 3.11. In Vivo Analysis

Transformed and non-transformed *Saccharomyces cerevisiae* strains were cultivated in YPD medium for 8 h at 30 °C, starting with an OD_600_ of 0.4. In order to overexpress Asf1 or Asf1^170-279^ proteins, the medium was supplemented with 2% galactose. OD_600_ values were measured after 2, 4, 6, and 8 h.

### 3.12. Western Blot Analysis

Yeast cells were collected after 8 h and disrupted by sonication. Lysates (25 µg protein) were separated on a 12.5% polyacrylamide gel. Afterwards, proteins were transferred onto a PVDF membrane. The His-tagged Asf1 protein was detected using an anti-His antibody conjugated with alkaline phosphatase (Merck KGaA, Darmstadt, Germany).

### 3.13. Statistical Analysis and Management Data

All data are expressed as mean ± SD (standard deviation) of three independent experiments and were analyzed using a *t* test with Statistica software 12.0. A *p* value < 0.05 was considered statistically significant.

### 3.14. CABS-Dock Method

This tool can be used in the docking analysis of peptides to proteins. The advantage of this protein–peptide docking is the opportunity for the flexible docking of a peptide to the kinase without any information about the peptide structure and/or the binding site. This step was performed with the CABS-dock online docking server [55,56,57], available at http://biocomp.chemuw.edu.pl/CABSdock/ (accessed on 2 October 2022). The modeled and docked structures were then analyzed with the UCSF Chimera 1.12rc software. The binding sites of the structures were identified using the crystal structures of CK2α (PDB code 1PJK), CK2α’ (PDB code 3OFM), and Asf1^170-193^.

Up to 40% of the interactions between proteins in higher eukaryotes are estimated to be mediated by peptidic binding motifs. These motifs are often located within locally unstructured protein regions, but can also bind their partners as short isolated peptides [58].

## 4. Conclusions

In this report, we showed that the Asf1 protein exhibited a weaker effect on holoenzymes compared to free catalytic subunits and caused an increase in the catalytic activity of the CK2α subunit in vitro, which may indicate various possibilities for the regulatory mechanisms of these isoforms. Results of the interaction between Asf1 and CK2 seemed to confirm the structural and functional diversity of different molecular forms of this protein kinase.

The different effects of Asf1 towards CKα and CK2α’ correlate with data obtained from inhibitory studies using halogenated *1H*-benzimidazole and flavonoid derivatives. During the past few years, we have already described several compounds possessing different inhibitory activity towards CK2 subunits [44,45,46,47,59]. Such compounds may help to improve the understanding of differences in the regulation and functions of both catalytic subunits.

## Figures and Tables

**Figure 1 ijms-23-15764-f001:**
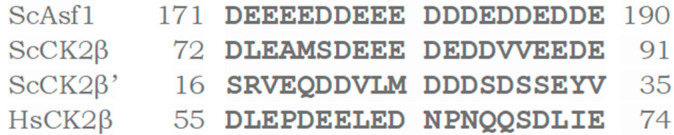
Sequence alignment of yeast Asf1 aa 171-190, yeast (Sc) CK2β aa 72-91, yeast (Sc) CK2β’ aa 16-35, and human (Hs) CK2β aa 55-74.

**Figure 2 ijms-23-15764-f002:**
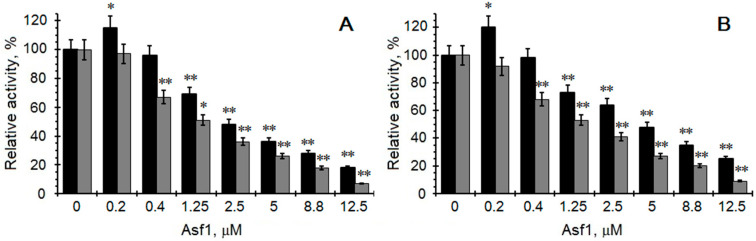
Effect of different Asf1 concentrations on kinase activity of CK2α (black bars) and CK2α’ (grey bars) from human (**A**) and yeast (**B**). The reaction was carried out using yeast P2B (10 µM) as a substrate and 20 µM ATP. Relative activities are calculated towards enzyme activity in the absence of Asf1. * and ** indicate significant difference compared to the controls (*p <* 0.05, *p <* 0.01).

**Figure 3 ijms-23-15764-f003:**
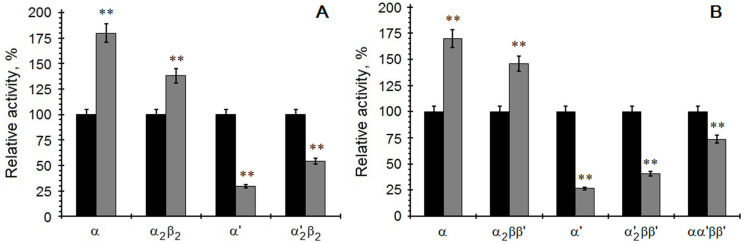
Kinase activity of different CK2 isoforms from human (**A**) and yeast (**B**) in the presence of Asf1. Comparison of activity in the absence (black bars) and in the presence of 1 µM Asf1 protein (grey bars) after preincubation. Relative activity is calculated towards enzyme activity in the absence of Asf1. ** indicates significant difference compared to the controls (*p <* 0.01).

**Figure 4 ijms-23-15764-f004:**
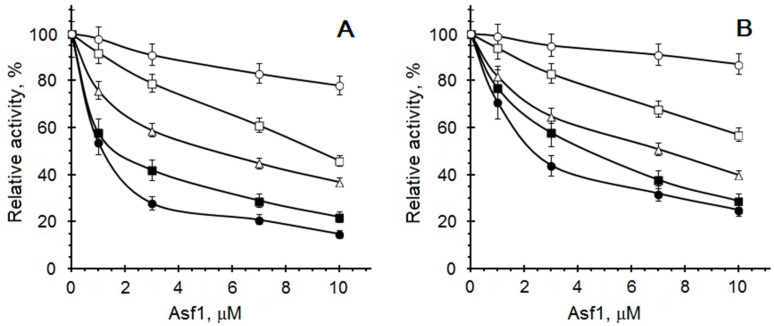
Comparison of substrate phosphorylation by CK2α’ (**A**) and CK2α (**B**) in the presence of different Asf1 concentrations. Enzyme activity was measured using the following protein substrates: Elf1 (∆), Fip1 (■), Svf1 (□), synthetic peptide (○), and P2B (●). The experiment was carried out in the presence of 10 μM of protein substrate and 20 μM ATP. Relative activity is calculated towards enzyme activity in the absence of Asf1.

**Figure 5 ijms-23-15764-f005:**
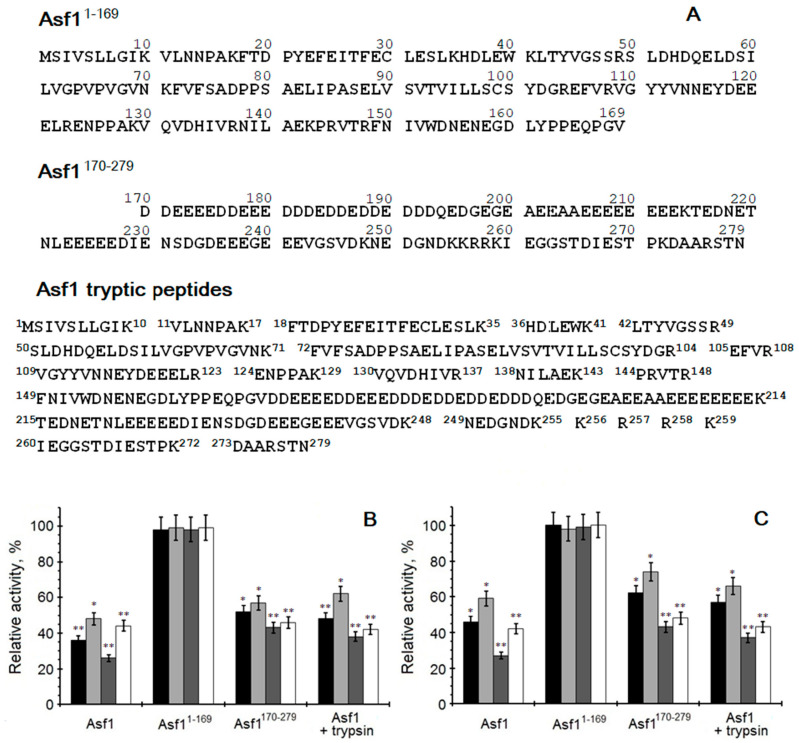
Comparison of effect of different Asf1 proteins/peptides (**A**) on CK2 activities from human (**B**) and yeast (**C**). Phosphorylation activity of CK2α (black bars), human CK2α_2_β_2_ or yeast CK2α_2_ββ’ (light grey bars), and CK2α’ (dark grey bars) and human CK2α’_2_β_2_ or yeast CK2α’_2_ββ’ (white bars), was measured in the presence of Asf1 or its fragments. The relative activity is given in reference to CK2 incubated with trypsin and STI as control. The experiment was carried out in the presence of 10 μM P2B and 20 μM ATP. * and ** indicate significant difference compared to the controls (*p* < 0.05, *p* < 0.01).

**Figure 6 ijms-23-15764-f006:**
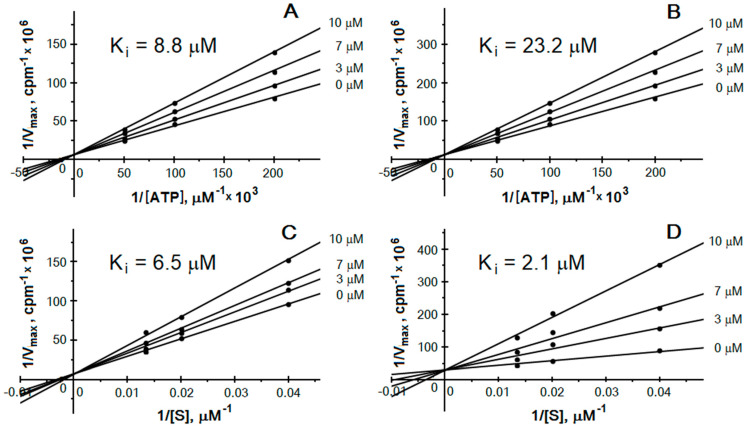
Inhibition of CK2 by Asf1. Lineweaver–Burk double reciprocal plots of CK2α (**A**,**B**) and CK2α’ (**C**,**D**) inhibition obtained in the absence and in the presence of the indicated full-length Asf1 (**A**,**C**) and Asf1^170-79^ (**B**,**D**) concentrations.

**Figure 7 ijms-23-15764-f007:**
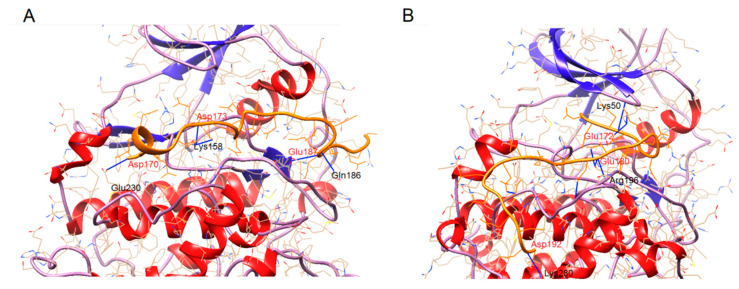
CABS-dock model of the binding modes. (**A**) CK2α (1PJK) and Asf1^170-193^, (**B**) CK2α’ (3OFM) and Asf1^170-193^. The peptide is shown in orange. Amino acids involved in the binding are given in black (enzyme) and red (peptide).

**Figure 8 ijms-23-15764-f008:**
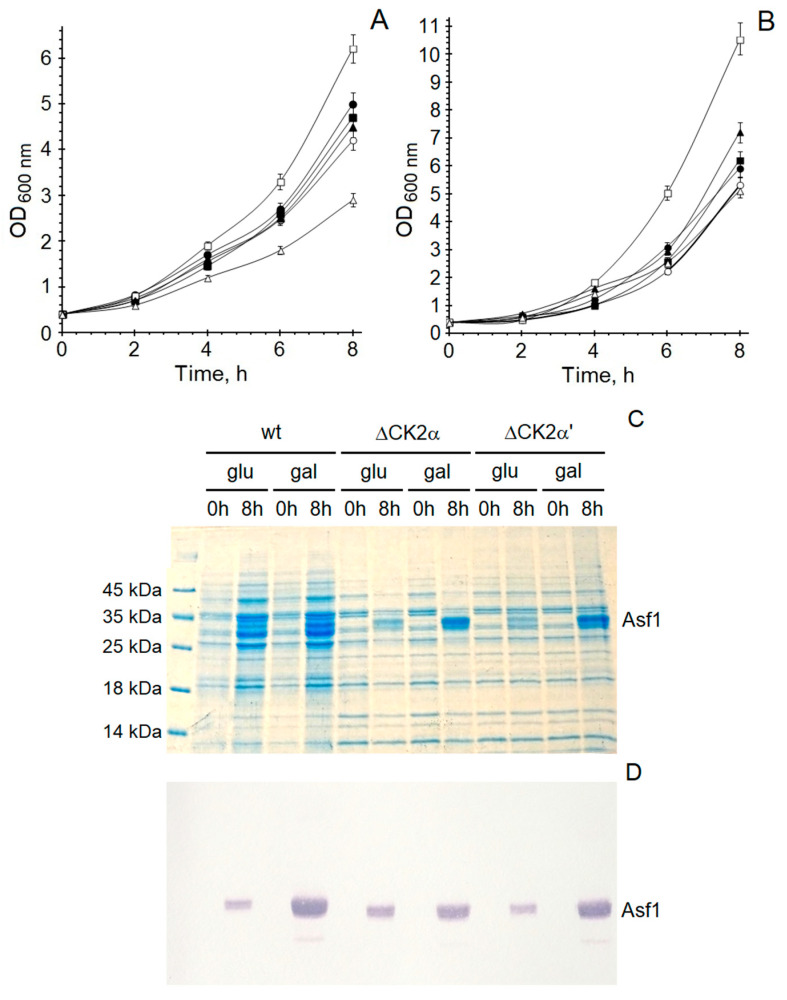
Influence of Asf1 protein towards yeast growth rate. (**A**) Medium containing galactose, (**B**) medium containing glucose. Optical densities determined after 0–8 h for cultures *S. cerevisiae* wt (●), *S. cerevisiae* wt/pYES2/CT::*Asf1* (○), *S. cerevisiae* ΔCK2α’ (■), *S. cerevisiae* ΔCK2α’/pYES2/CT::*Asf1* (□), *S. cerevisiae* ΔCK2α (▲), and *S. cerevisiae* ΔCK2α/pYES2/CT::*Asf1* (∆). (**C**) SDS/PAGE of lysates from 8 h cultures and (**D**) Western blot analysis of lysates from 8 h cultures.

**Table 1 ijms-23-15764-t001:** List of hydrogen bonds formed between CK2 catalytic subunits and Asf1^170-193^.

CK2α	Asf1 ^170-193^	Distance (Å)	CK2α’	Asf1 ^170-193^	Distance (Å)
K158	D177	3.553	K50	E174	3.291
K158	E180	3.295	K50	E175	3.061
E180	D189	2.938	K50	E178	3.141
Q186	E187	3.302	S195	E180	2.826
E230	D170	2.747	S195	E180	2.845
			R196	E180	3.362
			E231	D186	3.084
			G236	D181	2.983
			K280	D192	3.249

**Table 2 ijms-23-15764-t002:** Primer pairs used in this study.

Protein	Expression Vector	Primer Sequences *
Asf1	pET28a	5′-CCGGATCCTCAATTGTTTCACTGTTAGGCA-3′ 5′-ACGTCGACTTAATTCGTTGAACGTGCCGCA-3′
Asf1^1-169^	pET28a	5′-CCGGATCCTCAATTGTTTCACTGTTAGGCA-3′ 5′-CGGTCGACTTATAC GCCGGGCTGTTCAG-3′
Asf1^170-279^	pET28a	5′-ACGGATCCGATGATGAAGAGGAGGAGGACG-3′ 5′-ACGTCGACTTAATTCGTTGAACGTGCCGCA-3′
Asf1	pYES2/CT	5′-CCAAGCTTGGAAAAATGTCAATTGTTTCACTG-3′ 5′-CCGAATTCTTAATTCGTTGAACGTGCCGC-3′
Asf1^170-279^	pYES2/CT	5′-CCAAGCTTGGAAAAATGGATGATGAAGAGGAG-3′ 5′-CCGAATTCTTAATTCGTTGAACGTGCCGC-3′

* Restriction sites are underlined.

## Data Availability

The datasets generated for this study are available on request to the corresponding author.

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
