# Peer review of "Yeast Protein Asf1 Possesses Modulating Activity towards Protein Kinase CK2"

_ijms, 2022, doi:10.3390/ijms232415764_

Round 1
Reviewer 1 Report
Baier et al. investigate the effect of the nucleosome assembly factor Asf1, and in particular of its acidic C-terminal cluster, on the activity of protein kinase CK2. The effect on the different isoforms of the enzymes are shown, with very good and strong biochemical data. These data have interesting potential outcomes in the field of CK2 research.
However, to improve the paper, I have some comments:
-In Fig 1 the authors show an alignment considering yeast CK2 beta subunit. In yeast another regulatory subunit is present (CK2 β’). It would be interesting to know if this regulatory region is present on both regulatory subunits or just on the β one.
-In Fig 3 the CK2 isoform α’2ββ is shown. As far as I know, as reported in literature (Kubinski et al., 2007, Tripodi et al., 2011), an heterodimer ββ’ is required to form the CK2 holoenzyme, at least in vivo in yeast cells. Could you please explain this data? Are you sure that you are measuring the activity of a holoenzyme α’2ββ and not instead of free α’ catalytic subunit? The effect of Asf1 in both cases is similar. I have serious doubts on this specific data.
-In all figures, * indicating statistical significance should be added.
-Figure 8: the name of the strains should be easier to understand for the reader, such as wt, cka1delta, cka2delta and so on. Moreover, the authors mention two different media (with glucose or galactose, to induce overexpression), but just one curve is shown in Fig. 8. Both curves, in glucose and in galactose, must be shown, to confirm that the effects on growth rate are due to the overexpression of Asf1. In addition, a western showing Asf1 expression level in these strains, in glucose and in ethanol, should be added.
Minor issues:
-Although deletion of β subunits in yeast does not affect viability, as properly stated, it affects cell cycle progression through the G1/S transition (Tripodi et al, 2013) and the author should mention it in the introduction.
-before showing fig 5, I would suggest adding a small scheme showing the different Asf1 parts used, it would be very useful for the reader.
-some typos are present here and there in the text, please check carefully.
Author Response
List of responses to each of the Reviewer #1 comments
We appreciate Reviewer’s suggestions. They are unbelievably valuable for our manuscript.
In Fig 1 the authors show an alignment considering yeast CK2 beta subunit. In yeast another regulatory subunit is present (CK2 β’). It would be interesting to know if this regulatory region is present on both regulatory subunits or just on the β one.
Figure 1 was changed and the respective sequence in CK2β’ was added.
In Fig 3 the CK2 isoform α’2ββ is shown. As far as I know, as reported in literature (Kubinski et al., 2007, Tripodi et al., 2011), an heterodimer ββ’ is required to form the CK2 holoenzyme, at least in vivo in yeast cells. Could you please explain this data? Are you sure that you are measuring the activity of a holoenzyme α’2ββ and not instead of free α’ catalytic subunit? The effect of Asf1 in both cases is similar. I have serious doubts on this specific data.
This was a mistake in the figure and is now corrected. Of course the holoenzyme is α’2ββ’.
In all figures, * indicating statistical significance should be added.
Statistical significance was added to the figures.
Figure 8: the name of the strains should be easier to understand for the reader, such as wt, cka1delta, cka2delta and so on. Moreover, the authors mention two different media (with glucose or galactose, to induce overexpression), but just one curve is shown in Fig. 8. Both curves, in glucose and in galactose, must be shown, to confirm that the effects on growth rate are due to the overexpression of Asf1. In addition, a western showing Asf1 expression level in these strains, in glucose and in ethanol, should be added.
The names of the strains were simplified in the text and in figure 8 as suggested by the reviewer. The result (as figures) of the western blot was added.
Although deletion of β subunits in yeast does not affect viability, as properly stated, it affects cell cycle progression through the G1/S transition (Tripodi et al, 2013) and the author should mention it in the introduction.
The publication was added in the introduction.
before showing fig 5, I would suggest adding a small scheme showing the different Asf1 parts used, it would be very useful for the reader.
A scheme showing the different Asf1 proteins used in this study was added to figure 5.
some typos are present here and there in the text, please check carefully.
The manuscript was read again carefully and corrected.

Reviewer 2 Report
The manuscript on CK2 activation/inhibition complements the existing information about the possible mechanisms of action of such an important protein.
Some points need to be clarified for a better understanding.
Figure 3.
The CK2 subunits and holoenzymes in humans and yeast in graphs A and B are different. Explain in detail why they were not used. Advantages and disadvantages of interpretation.
Figure 4
Why were CX-4945 and/or TBB not used to compare the inhibitory effects?
What type of CK2 inhibitory peptides are expected to be generated by trypsin hydrolysis?
What convenience is there to test another type of protease to obtain peptides with more adequate sequences to increase inhibition?
Figure 5
Explain why intact Asf1 induces less relative activity than Asf1 fragment 1-169?
Figure 6
Place the kinetic constants, as well as the equations obtained and expand the discussion.
Author Response
List of responses to each of the Reviewer #2 comments
We appreciate Reviewer’s suggestions. They are unbelievably valuable for our manuscript.
Figure 3.
The CK2 subunits and holoenzymes in humans and yeast in graphs A and B are different. Explain in detail why they were not used. Advantages and disadvantages of interpretation.
We are not sure how to understand this comment. CK2 holoenzymes differ between human and yeast due to the fact that human CK2 is built of only one regulatory subunit, whereas yeast possesses two of them.
Figure 4
Why were CX-4945 and/or TBB not used to compare the inhibitory effects?
The effect of TBB was already shown in one of our former publications (Janeczko et al., 2011). This information was added to the manuscript.
What type of CK2 inhibitory peptides are expected to be generated by trypsin hydrolysis?
The expected peptides after trypsination were added are shown in figure 5A.
What convenience is there to test another type of protease to obtain peptides with more adequate sequences to increase inhibition?
Using trypsin was only a preliminary experiment to analyse if shorter peptides also possess effects on CK2 activity. Afterwards, the Asf1 fragments 1-169 and 170-279 were produced for further experiments.
Figure 5
Explain why intact Asf1 induces less relative activity than Asf1 fragment 1-169?
Asf1 aa 1-169 did not contain the acidic stretch which is responsible for the inhibition of CK2. This explanation is already given in the text.
Figure 6
Place the kinetic constants, as well as the equations obtained and expand the discussion.
The kinetic constants were added.

Round 2
Reviewer 1 Report
The authors have answered to all my questions. In my opinion the paper is very good and deserves to be published.